# A network property necessary for concentration robustness

Jeanne M.O. Eloundou-Mbebi[1], Anika Küken[1], Nooshin Omranian[1], Sabrina Kleessen[2], Jost Neigenfind[2], Georg Basler[3] & Zoran Nikoloski[1]

Maintenance of functionality of complex cellular networks and entire organisms exposed to environmental perturbations often depends on concentration robustness of the underlying components. Yet, the reasons and consequences of concentration robustness in large-scale cellular networks remain largely unknown. Here, we derive a necessary condition for concentration robustness based only on the structure of networks endowed with mass action kinetics. The structural condition can be used to design targeted experiments to study concentration robustness. We show that metabolites satisfying the necessary condition are present in metabolic networks from diverse species, suggesting prevalence of this property across kingdoms of life. We also demonstrate that our predictions about concentration robustness of energy-related metabolites are in line with experimental evidence from *Escherichia coli*. The necessary condition is applicable to mass action biological systems of arbitrary size, and will enable understanding the implications of concentration robustness in genetic engineering strategies and medical applications.

[1] Systems Biology and Mathematical Modeling Group, Max Planck Institute of Molecular Plant Physiology, Am Muehlenber 1, 14476 Potsdam-Golm, Germany. [2] Targenomix, Am Muehlenberg 11, 14476 Potsdam-Golm, Germany. [3] Department of Chemical and Biomolecular Engineering, University of California, Berkeley, California 94720, USA. Correspondence and requests for materials should be addressed to Z.N. (email: nikoloski@mpimp-golm.mpg.de).

Robustness against environmental fluctuations is found across different scales of cellular organization, from metabolite levels and enzyme activities[1–5] to complex cellular functions[6–10]. Cellular components (for example, transcripts, proteins and metabolites) involved in regulation and control of physiological functions do not act in isolation, but form intricate cellular networks. Therefore, understanding how robustness of concentrations arises in the context of these large-scale biochemical networks can point to mechanisms that allow maintenance of cellular functions in a narrow range. Further, pinpointing the reasons and consequences of concentration robustness can help us elucidate the strategies employed by single cells and entire organisms to mitigate the effects of intracellular and environmental perturbations[11,12].

Identifying cellular components that exhibit concentration robustness in a biological system exposed to environmental perturbations is a non-trivial task. To this end, the (steady-state) concentration for a given cellular component is determined from samples of genetically identical organisms subject to different environments. These measurements are based on various resource-intensive phenotyping technologies[13–15]. For instance, identification of metabolites and measurement of their cellular concentrations with modern metabolomics technologies necessitates the usage of expensive authentic chemical standards or labelling techniques[15–17]. Moreover, the existence of cellular components whose pools are partitioned among various cellular compartments[18] further complicates the study of concentration robustness. It may be the case that a compartment-specific concentration is robust to environmental perturbations, although the entire component pool may not be maintained in a narrow range. Therefore, more elaborate experiments often have to be designed to allow for extraction of subcellular pools of cellular components[19,20]. As a result, the model-driven identification of concentration robustness on a subcellular level will greatly benefit from methods that allow selection of cellular components for targeted experiments.

Recent systems biology efforts have resulted in the assembly of large-scale models that consider the entirety of known biochemical reactions at various levels of cellular organization, from gene regulation and signalling to metabolism[21]. Therefore, one promising possibility to design targeted experiments with the aim of identifying components with robust concentrations is to rely on the analysis of these mechanistic network-based descriptions of cellular activities.

Given a biochemical network model, here we identify a network-based condition which must be satisfied by each cellular component exhibiting concentration robustness to environmental perturbations. Therefore, any component violating the identified structural condition can be excluded from further experimental investigation of robustness to environmental perturbations. We show that the identified structural property underlying the condition necessary for concentration robustness can be efficiently determined for genome-scale metabolic networks. As a result, we use the derived necessary condition to test the possibility for prevalence of concentration robustness for metabolites across different organisms under realistic modelling assumptions. In addition, we examine the effect of network perturbations on our findings, and show that the predictions about lack of concentration robustness are in line with experimental evidence and kinetic modelling of *Escherichia coli*'s metabolism.

## Results

**Network concepts to study concentration robustness.** A biochemical network exhibits absolute concentration robustness (ACR) for a given cellular component if the concentration of the component is the same in every positive steady state[2]. It has been shown that existence of components with ACR allows for sustaining normal cellular function (for example, growth) under suboptimal conditions[4]. In the following, we define and illustrate the key concepts which are essential for deriving the necessary

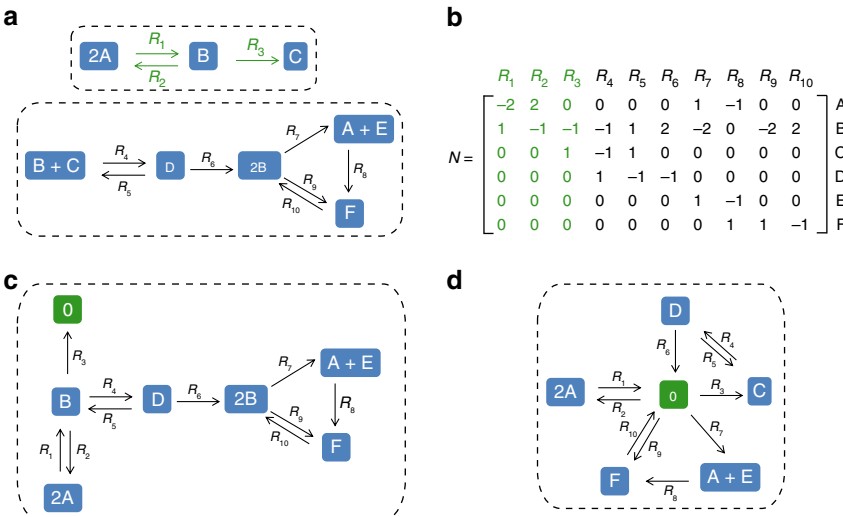

**Figure 1 | Illustration of the network concepts and the derived necessary condition for concentration robustness.** (**a**) Standard reaction diagram of a network in which six components, A–F, are interconverted by 10 reactions. The reaction diagram has $n = 8$ nodes, corresponding to complexes and 10 edges, representing the reactions. The two linkage classes are surrounded by dashed lines. (**b**) Stoichiometric matrix $N$ of the network in **a**. Reactions $R_1$, $R_2$ and $R_3$, with the corresponding edges in the reaction diagram and columns in $N$ coloured in green, belong to the same linkage class (**c**) Standard reaction diagram for the network in **a** upon removal of component C. Since C exists as a single-component complex in **a**, its removal introduces the zero complex, O, coloured in green. This network contains $n = 7$ nodes, 10 reactions, $l = 1$ linkage class. (**d**) Standard reaction diagram for the network in **a** upon removal of component B. Since B exists as a single-component complex in **a**, its removal introduces the zero complex, 0, coloured in green. The network in **a** upon removal of B contains $n = 6$ nodes, 10 reactions, $l = 1$ linkage class. The structural deficiencies of the networks in **a**,**b** are $\delta_s = 1$, while for the network **c**, $\delta_s = 0$.

structural condition for ACR—stoichiometric matrix[22] and standard reaction diagram[23]

A biochemical reaction is specified by two positive linear combinations of components, called substrate and product complexes, respectively. A reaction vector is given by the difference of the product and substrate complexes of the corresponding reaction. For a set of biochemical reactions, the stoichiometric matrix, $N$, comprises the reaction vectors as its columns and rows that correspond to the components. For instance, the stoichiometric matrix corresponding to the reactions in Fig. 1a is given in Fig. 1b. Here, every reversible reaction is split into the forward and backward irreversible reactions.

A standard reaction diagram then corresponds to the directed graph whose nodes represent the complexes, and a directed edge connects two nodes if they refer to the substrate and the product complex of a reaction, respectively. For instance, a standard reaction diagram with 10 irreversible reactions, represented by directed edges, and eight complexes (that is, nodes) composed of six components, A–F, is shown in Fig. 1a; B + C is the substrate complex and D is the product complex of reaction $R_4$.

The set of nodes connected by paths in the reaction diagram belong to the same linkage class[24]. For instance, in Fig. 1a, complex 2A is in the same linkage class (marked by green edges) with complex C as they are connected by a path; in contrast, the complexes 2A and 2B do not belong to the same linkage class. Therefore, the number of linkage classes corresponds to the number of connected pieces in a given standard reaction diagram. The network in Fig. 1a has two linkage classes ($l = 2$) surrounded by dashed lines.

With the help of these concepts, we present the notion of structural deficiency of a network, denoted by $\delta_s$, central to the chemical reaction network theory[25]. The structural deficiency of a network is given by $\delta_s = n - l - r(N)$, where $n$ is the number of nodes, $l$ is the number of linkage classes and $r(N)$ is the rank of the stoichiometric matrix (that is, the maximum number of linearly independent reaction vectors, which can be efficiently computed with standard techniques[26]). Therefore, the structural deficiency can be efficiently determined for a reaction network of arbitrary size. It is known that for any network the structural deficiency is a non-negative integer which has been associated to the existence and uniqueness of steady-states[25]. For the network in Fig. 1a, there are $n = 8$ complexes, $l = 2$ linkage classes, the rank of the stoichiometric matrix is $r(N) = 5$. Hence, the structural deficiency is $\delta_s = 8 - 2 - 5 = 1$.

### Absolute concentration robustness in mass action networks.
The change in concentration of a given component is shaped by the stoichiometry and the rates of biochemical reactions in which the component participates as a substrate or a product. The property of ACR has already been extensively studied for networks of reactions whose rates are described by the widely used mass action kinetics. In real-world applications, each component is associated with a positive mass. A network of biochemical reactions is termed conservative, if the masses of the substrate and product complexes of each reaction are the same[27]. It is known that conservative mass action networks of deficiency zero cannot contain a component exhibiting ACR irrespective of the values assigned to the rates constants[28].

For mass action networks, there also exist structural properties that provide sufficient conditions for existence of a component with ACR[29,30]. However, these sufficient conditions are either too restrictive or cannot be efficiently computed in large-scale metabolic networks typically employed in studies of metabolism[31–33]. For instance, one of the sufficient conditions can only be invoked for networks of structural deficiency of one[2].

However, genome-scale metabolic networks are of considerably larger deficiency (Supplementary Table 2). The other sufficient condition treats the rate constants as symbols[29] and relies on determining invariant linear combination of complexes. Despite the advances in symbolic computation, systematic determination of such invariants becomes computationally infeasible for real-world large-scale metabolic networks[29]. Therefore, determining a structural condition necessary for ACR of a particular component offers another alternative to analyse concentration robustness in large-scale networks.

### Structural deficiency and absolute concentration robustness.
Our main result is based on establishing whether or not the structural deficiency changes upon removing a single component from the network. To this end, we rely on the network obtained by eliminating a given component from each complex containing the component. Removal of a component may drastically alter the network, in terms of number of nodes, linkage classes and the rank of the stoichiometric matrix. For instance, the network in Fig. 1a upon removal of component C is illustrated in Fig. 1c; it has $n = 7$ nodes, $l = 1$ linkage class, and the rank of the stoichiometric matrix is $r(N) = 5$; the structural deficiency is, thus, of value $\delta_s = 7 - 1 - 5 = 1$. The number of complexes is reduced in the new system since the complex B + C upon removal of C coincide with the complex B, present in the original network. This is the reason why the stitching of the reactions is changed, leading to a single linkage class. In addition, removal of a component may lead to the introduction of the so-called zero complex[34]. This is the case upon removal of components B from the network in Fig. 1a (Fig. 1d).

The idea of removing a component from biochemical reaction network has been previously employed to make statements about the possibility of the network to exhibit multistationarity[35]. Namely, for a given set of rate constants, it has been shown that if a reaction system obtained upon removal of a component admits multiple non-degenerated positive steady states, so does the original system. Therefore, this result may be used to identify subnetworks conferring multistationarity to the entire network. Here, we establish a connection between a structural deficiency, as a key network invariant, and ACR for a particular component. It is this connection that allows us to apply the results to large-scale networks, typically arising in the study of metabolism.

We now have the concepts required for stating our main result, proved in Supplementary Methods.

**Theorem**. Consider a mass action reaction system that for given rate constants admits a positive steady state with and without removal of a component S. If the system has ACR in species S, then the systems with and without removal of S have the same structural deficiencies.

The removal of a component assumed to exhibit ACR in a mass action system can be intuitively understood as rescaling of the rate constants for the reactions in which the component appears as a substrate. This allows us to establish a correspondence between the two reaction systems with respect to the linear combinations of reaction rates around each complex, leading to our theoretical result.

Our necessary condition can be used to pinpoint the components which do not show ACR. Therefore, it can be readily employed to reduce the number of components for which targeted experiments over multiple environments must be planned to explore and validate the possibility for ACR. For instance, component B in the network on Fig. 1a does not have ACR since the structural deficiencies of the original and of the network in panel a upon removal of B differ (Fig. 1d). In contrast,

the structural deficiencies of the original network and the network upon removal of component C coincide (Fig. 1c), and C cannot be precluded from having ACR. Analytic solution demonstrates that, indeed, component C does exhibit ACR. However, although removal of the components A, D, E and F does not alter the deficiency, these components do not exhibit ACR (as shown in the analytical solution in Supplementary Methods). As already pointed out, our theorem is not applicable to conservative mass action networks of deficiency zero in which the existence of ACR components is precluded[28].

**Applications to metabolic networks.** Our mathematical results provide a tractable way to restrict the number of metabolites to be tested for ACR in genome-scale metabolic networks, under the assumption that these networks support positive steady states. To this end, we analyse 14 networks from a broad range of organisms spanning all kingdoms of life, which vary greatly in their sizes, tissues, subcellular compartmentalization and uni- or multicellular organization (Supplementary Table 1). We found significant correlations of 0.89 ($P$ value $= 2.17 \times 10^{-5}$, Fisher $z$-transformation, $R^2 = 0.79$) between the number of metabolites with potential for ACR and the total number of metabolites (Fig. 2). The implication of this finding is that the number of metabolites to be tested for ACR can be reduced by, on average, 42.5% of the total number of metabolites over the investigated species (Supplementary Table 2).

We also conduct a simulation study of a well-investigated medium-size kinetic model of *E. coli* consisting of 830 components (that is, metabolites and enzymes) and 1,330 reactions. The rates of these reactions are modelled with mass action kinetics and are associated positive rate constants that yield a positive steady state from an initial condition[36], thus satisfying the hypothesis of our theoretical result. Altogether, 784 components do not violate the necessary condition and, thus, may exhibit ACR. To narrow down the search for components that are likely ACR in this model, we simulate 150 different positive steady-state concentrations starting from 150 different initial conditions. By analysing the identified steady-state concentrations, we identify two components whose steady-state

concentration is unchanged upon perturbations in initial conditions (Supplementary Table 3). Therefore, only 0.26% of components that satisfy the necessary condition in this model are likely to show ACR, although simulation studies cannot provide a conclusive answer.

We next focus on identifying whether the necessary condition for ACR holds for compounds essential for characterizing the energy status of biological systems, namely, the oxidized and reduced version of NAD and NADP as well as the adenosine phosphates (that is, AMP, ADP and ATP)[37]. These compounds provide the energy for driving the biochemical reactions in which they participate. Under the assumption that the networks support positive steady states with mass action kinetics, we find that in the networks of archaea and bacteria generally these compounds violate the necessary condition (Supplementary Table 2), in line with experimental observations[38,39]. This result suggests that simple organisms may have not evolved mechanisms to maintain specific levels of energy-related metabolites. Our results can also be used to base simulation studies of metabolic networks on more appropriate biochemical assumptions, since energy-related compounds in such studies are often assumed to be constant. Moreover, while these compounds do not show ACR in the genome-scale network of *E. coli*, NADH, NAD, NADP and ATP satisfy the necessary condition in the highly simplified core metabolic network used for estimating fluxes from labelling experiments. Therefore, our results point out that the necessary condition for ACR in selected subnetworks may not match the predictions based on the entire metabolic network. Hence, ACR is a truly systemic property which arises from the network as a whole and may not be conferred by its presence in the network modules. Our prediction that ATP does not show ACR in *E. coli* is in line with recent experimental evidence about variability of ATP concentration in single cells[40].

In contrast to results pertaining to archaea, bacteria and fungi, the metabolic networks of plants and animals contain compounds characterizing the energetic status that satisfy our necessary condition for ACR. Specifically, our necessary condition applied on the metabolic networks of *Chlamydomonas reinhardtii* is satisfied for NADP and ATP in more than one cellular compartment. This result indicates that, even if the total pool

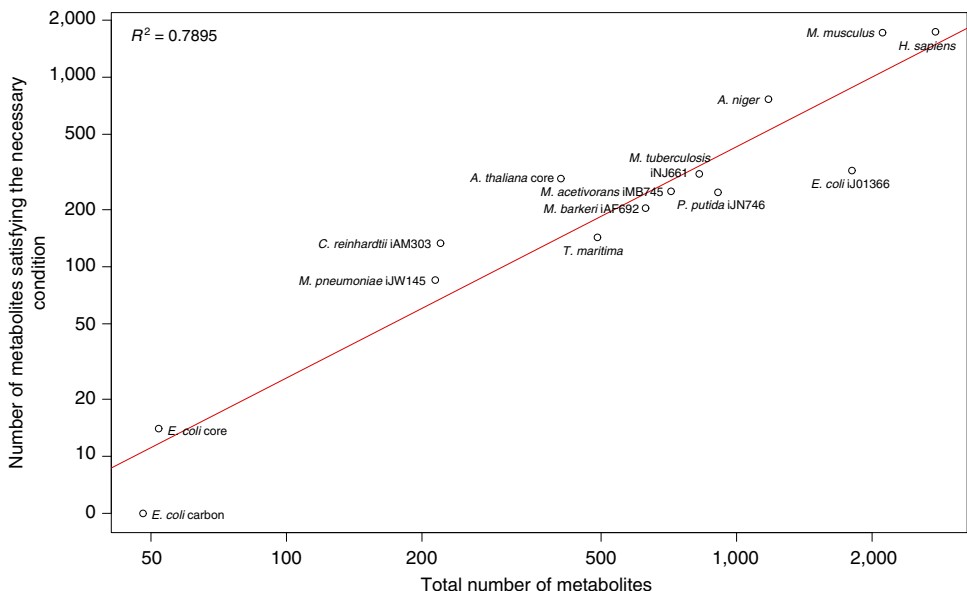

**Figure 2 | Prevalence of concentration robustness across kingdoms of life.** Log–log plot of the number of metabolites satisfying the necessary condition for ACR as a function of the total number of metabolites. The metabolic networks are from species across all domains of life from archaea, bacteria and fungi to plants and animals.

of a metabolite, on the level of the entire cell, varies with environmental changes, its subcellular concentration may in fact exhibit ACR. Therefore, our findings provide important direction for experimental planning and conducting measurements of subcellular concentrations for specific metabolites. For instance, the oxidized and reduced version of NAD and NADP do not violate the necessary condition in most of the compartments (Supplementary Table 2), in line with modelling evidence[41].

**Effects of network perturbations**. The analysed large-scale metabolic networks include the characterized enzymatic activities in different organisms. However, due to the existence of uncharacterized gene functions, some of these networks may be incomplete. Therefore, to examine the sensitivity of our findings to this bias in the network, we consider removing 1, 2, 5, 7 and 10% randomly selected reactions from each of the analysed networks. To quantify the effect of the reaction removal, we focus on those metabolites that satisfy the necessary condition in the original network (Supplementary Table 2 for the number of metabolites satisfying the necessary condition). For these metabolites, we identify those which violate the condition upon the network perturbation. We refer to such metabolites as switching metabolites. We then determine the switching ratio, defined as the proportion of switching metabolites from those which satisfy the necessary condition in the original network. The switching ratio for every network and perturbation level (that is, percentage of removed reactions) was determined over at least 40 samples.

Our results indicate that the switching ratio depends on a small set of reactions (Fig. 3, Supplementary Fig. 1). This is supported by the observation that already the removal of 1% of reactions, on average, leads to as large value for the switching ratio as the removal of 10% of the reactions. In addition, since the switching ratio is not larger than 0.5 across all networks (with exception to *Chlamydomonas reinhardtii*), we conclude that some metabolites satisfy the necessary condition even upon all considered levels of perturbations. This finding suggests that these metabolites may

essentially participate in pathways which may be effectively decoupled in the considered networks.

Another source of uncertainty of large-scale metabolic networks is represented by the directionality of the included reactions. While some reactions are known to operate effectively as irreversible, others may change the operating direction preferentially according to cellular conditions[42]. Nevertheless, changing the directionality of a reaction does not affect the number of complexes and the number of linkage classes. Moreover, the rank of the stoichiometric matrix is not affected by change of directionality. These facts together with the definition of structural deficiency demonstrate that change of reaction directionality does not affect the structural deficiency. Therefore, our findings are not affected by possible uncertainty in reaction directionality.

## Discussion

Our necessary condition for ACR is applicable to any mass action network irrespective of the values of the rate constants ensuring positive steady states. Since Michaelis–Menten kinetics is derived from mass action, the derived necessary conditions can also be used to rule out the possibility of ACR for components in systems endowed with this type of kinetics. Our simulations studies, however, indicate that the number of components that could likely exhibit ACR may be substantially lower than that implied by the necessary condition. Nevertheless, the tractable means for precluding ACR in combination with genome-scale metabolic networks can be used for model-driven planning of experiments under different environments and with variety of organisms. Altogether, our findings pave the way for studying the evolutionary implications of ACR on a genome-scale level as well as the role of ACR in metabolic diseases and metabolic engineering strategies.

## Methods

**Large-scale metabolic network models**. We demonstrate the applicability of our approach on 14 metabolic networks from a broad range of organisms spanning

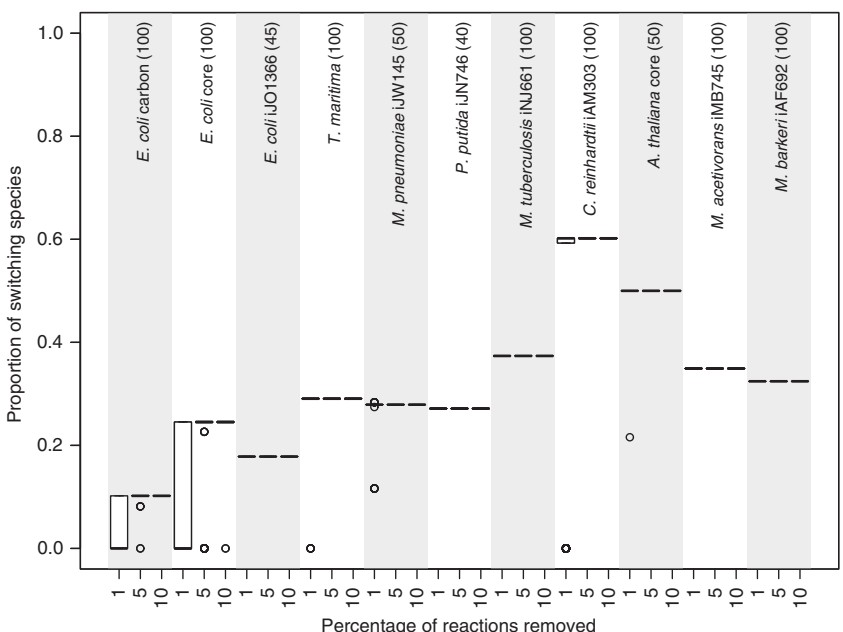

**Figure 3 | Effects of network alterations on the findings from applications of the derived necessary condition.** To investigate the effect of bias in the network, 1, 5 and 10% randomly selected reactions are removed from each of the analysed networks. The effect of the reaction removal is quantified by the proportion of switching metabolites. The switching ratio for every network and perturbation level was determined over at least 40 samples (indicated above the name of the networks analysed).

all kingdoms of life. These metabolic networks vary greatly in their sizes, tissues, subcellular compartmentalization and uni- or multi-cellular organization (Supplementary Table 1). The models are of high quality, as each considered reaction preserves mass and charge balance; therefore, they are mass conservative, as is our example in Fig. 1a. Reactions that do not carry flux in any steady state, so-called blocked reactions, are excluded from the analysed networks, as they preclude the existence of a positive steady state. In addition, each reversible reaction is split into two irreversible reactions (see Supplementary Table 2 for characteristics of the analysed models). To examine the sensitivity of our findings to network perturbations, we consider removing 1, 2, 5, 7 and 10% randomly selected reactions from each of the analysed networks.

**Structural deficiency.** To calculate the structural deficiency for a given metabolic network, we use a stand-alone application written in the statistical programming environment R version 3.2.1 based on functions in the igraph package[43]. To this end, we determine the number of complexes and the number of linkage classes in the reaction diagram derived from a given stoichiometric matrix, as well as the rank of the stoichiometric matrix specified by the model. The structural deficiency of a network upon removal of given species is determined based on a stoichiometric matrix from which the row corresponding to the species is removed.

**Kinetic modelling.** We employ a kinetic model of *E. coli*[36] from which we considered only the reactions associated to positive rate constants. The model was simulated from 150 different, randomly selected positive initial conditions, each leading to a different positive steady-state concentrations for the components. The model is simulated in MATLAB with the ode15s solver.

**Data availability.** All relevant data are available from the authors upon request.

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

## Acknowledgements

G.B. was supported by a Marie Curie Intra European Fellowship FP7, ERC grant # 329682ents. J.M.O.E.-M., A.K., N.O. and Z.N. are supported by the Max Planck Society.

## Author contributions

J.M.O.E.-M. derived the theoretical results and analysed data. A.K. analysed the stoichiometric and kinetic models. N.O., S.K. and G.B. analysed data and stoichiometric models. J.N. contributed to the theoretical results. Z.N. conceived and designed the study, and contributed to the theoretical results. All authors participated in writing the paper.

## Additional information

**Competing financial interests:** The authors declare no competing financial interests.

