## [Peer Review File · Nature Communications]

Reviewer #1 (Remarks to the Author)

Dear Editors,

The manuscript by Eloundou-Mbebi et al follows up on influential recent work by Shinar and Feinberg (among others) to establish structural conditions for absolute concentration robustness (ACR), a condition in which one protein in the network has the same steady state concentration regardless of the total amount of all proteins in the system. Rather than finding sufficient conditions for ACR, as has been the norm in the literature, they set out to find **necessary** structural conditions for this property. They find that in order for a protein to have ACR, its removal cannot change the deficiency of the network. The deficiency is an integer number that is independent of rate constants, i.e. determined exclusively by the network structure.

From a theoretical standpoint this is quite a nice result that complements existing work. As in the work by Shinar and Feinberg, the new condition can be easily determined by just writing the two networks and doing some simple computations in your head. This result could in principle be used to rule out ACR in given proteins assuming knowledge of the relevant network. The authors also carry out some analysis of several metabolic networks, showing that around 15% of the proteins in each system satisfy their necessary condition.

The idea of removing a species from a chemical reaction network is central to this work and it has been addressed in the literature before, see eg the paper by Joshi and Shiu (J Mathematical Chemistry 51(1):153-178, 2013). The authors should include this reference and how their ideas fit into this context.

My main concern with this paper is whether this is the appropriate venue for its publication. This result is not as readily applicable to biologists, since it doesn't actually establish ACR but only narrows possible ACR down to a smaller subset of proteins. Simple titration experiments can also rule out ACR by showing variability in steady state concentrations, even when the network is not known. Also, in my opinion the current quality of the writing and the figures, and the strength of the main result itself, are not up to the standards of Nature Communications.

I did like the theoretical aspects of this work quite a bit, which will be reflected in my comments and suggestions. My opinion is that the authors should submit to a more theoretical venue and work on the quality of the presentation as suggested. In particular, I suspect the main result can be significantly strengthened by eliminating assumptions such as positive deficiency or that deleting the ACR variable preserves the number of reactions. The main text has no structure eg introduction, background, main results, conclusion. It starts defining technical concepts on the second paragraph, which is too early. An introductory section should be expanded, including for instance what the system is robust to (eg with respect to what). A conclusion section should also be included.

Specific comments:

* page 2, top: 'which lies'

* page 2, top paragraph: A simple subfigure with a sample reaction and its corresponding stoichiometry matrix would clarify these concepts.

* page 2, 'The set of nodes which are...': This sentence is confusing - if I didn't already know what a linkage class is, I don't think I would understand from reading this.

* Main Theorem: Isn't the theorem true also for systems with zero structural deficiency? In that case the conclusion must be true, since the deficiency after eliminating S must also be zero. I didn't see that you use this assumption anywhere in the proof.

* Main Theorem: You assume that removal of S does not alter the number of reactions, to prevent reactions of the form $X \rightarrow X$. Such reactions are ruled out in chemical reaction networks, simply for the reason that they would not change the dynamics, eg the reaction vector is $X - X = 0$. I think this assumption is artificially eliminating candidates that could have the ACR property (or that maybe can never have it, you can discuss this too).

* page 3, bottom paragraph: what do you mean by 'characterizing the energy status of biological systems'?

* page 4, 'In contrast to archa...': In the first sentence of this paragraph you say that certain compounds in animals satisfy ACR necessary condition. In the next sentence you give as example molecules that do not satisfy ACR. Please clarify.

* page 4, bottom: 'The network upon removal' \rightarrow 'The network in panel (a) upon removal'

Supplementary Material Comments

* Author descriptions: Georg Basler is identified by number 3) in the main manuscript, here with 2).

* Def 1.1, 2) "of distinct vectors $y \in \dots$ "

* Def 1.1 3) "where $(y, y') \in R$ is denoted by $y \rightarrow y'$, and we say y 'reacts to' y' . Also, we assume that..."

* Def 1.4, bottom: c_y should be denoted c^y .

* Def 1.5: Are you assuming that the number of reactions is not changed upon removal of S? If so, you should state so explicitly throughout the manuscript.

* After (5): A_k is also a linear map, not non-linear as you state

* after (6): define w_y

* Remark 1.14: You can include a proof here after what you wrote above - it is obvious that $\text{Im } A_k \subseteq Q$, so $d_d = \dim \ker Y | \text{Im}(A) \leq \dim \ker Y | Q = d_s$

* page 7, item 2.: We show AN isomorphism between...

* page 7, last displayed equation: this relation notation seems awkward - why not just label the species S_1 through S_m , and refer to them as S_i, S_j etc?

* page 7, before prop 2.1: Again, why nonzero deficiency? In the zero deficiency case it would appear that the theorem also holds, because the necessary condition is always satisfied.

* Proposition 2.1: This proposition is referred to as prop 3.1 everywhere below. Same problem with the lemmas etc. You should use floating proposition labels in case you change section

numbers

* Lemma 2.2: this statement is very unclear - you state that a certain network is 'a reaction system', and then you proceed to define rate constants in a very specific way. The rate constants you choose are your own choice, and they don't prove that the network is well defined. Instead of Lemma 2.2, you should write a definition of the rate constants as you want to do it, and then show that this definition has desired properties.

* equation (16): sum over y in C_{-S} ?

* page 9, top: What is the D here, presumably the result of deleting S ?

* eq (19): I don't understand this notation Y_S . You can without loss of generality assume that the ACR species is eg the first one S_1 or the last one S_m , and simplify the notation throughout.

* Proposition 2.6: "and such that removal of S doesn't change the number of reactions..."

* Lemma 2.7: why is it that you didn't need to deal with duplicated complexes in the previous analysis?

* Lemma 2.7, eq (29): script Q

* Lemma 2.7, after (29): "where M is the matrix that removes the rows... "

* Lemma 2.7: It is unclear if Q refers here to the set for the original reaction or for the reaction after removing S

* Theorem 2.8: you never seem to have used above that deficiency >0 for the original system. You did assume that removal of S did not alter reaction, but you didn't write this explicitly in the propositions.

Reviewer #2 (Remarks to the Author)

Report on "Network structure is necessary for concentration robustness"

Chemical Reaction Network Theory (CRNT), developed by Martin Feinberg and others, reveals the underlying "almost linear" structure of deterministic mass-action kinetics. This linearity sets strong constraints on this type of dynamical systems, which lead to the basic results of CRNT. The central results are structural criteria for stability and existence of steady-state fixed points of the networks, which follow from conditions on the dimensions of the underlying vector spaces. Recently, there has been new interest in CRNT in the context of biological networks. In particular, the works of Shinar et al. derived a sufficient condition for Absolute Concentration Robustness (ACR), the invariance of the concentration of certain chemical species at all feasible steady states of the network. ACR may be relevant to biological function as a mechanism for stabilizing the concentration of important chemical species against fluctuations.

In the present paper, the authors derive a necessary condition for ACR for a class of chemical networks, those where the removal of certain species S does not change the number of reactions. In this class, if the system exhibits ACR w.r.t. species S , it must have the same structural deficiency with and without the species S . The authors then apply their condition to 14 networks and find that the fraction of chemical species that may exhibit ACR according to the condition fluctuates around 15% (Fig. 2). They also find that certain central energy carriers, such as ATP and NADP, satisfy the ACR condition in certain organisms, whereas in other organisms the condition is violated. The authors thus show that the existence of ACR for important compounds is

not universal but specific to the organism. I can see the potential interest in publishing the present results, if the authors can address the following points:

1) How strong is the condition in real networks? In other words, can the authors estimate, e.g. by some random sampling:

a. What is the fraction of species that obey ACR but their omission changes the number of reactions, and are therefore outside the scope of the condition?

b. What is the fraction of species that obey the condition of invariance of deficiency and number of reactions, but do not exhibit ACR?

2) How general is the present necessary condition with respect to the previous sufficient condition of Shinar and Feinberg:

a. How many of the ACR cases observed by the authors would be missed by the previous condition?

b. How many of the ACR cases that obey the Shinar-Feinberg condition violate the present condition?

3) "... some of these properties cannot be efficiently computed for large-scale biochemical networks^{12,13}, usually employed in studies of metabolism¹⁵⁻¹⁷, as they depend on the rate constants. "

a. the authors should elaborate on what exactly are the properties that cannot be efficiently computed and why.

b. A basic concept of CRNT is that counting dimensions and calculating topological indices such as the deficiency, often provides a lot of information regarding fixed points etc. For example, the Shinar-Feinberg is purely structural. Which CRNT criteria "depend on the rate constants"?

c. Can the authors substantiate the claim that the other criteria "are too restrictive for application with real-world networks, as they hold for a very special class of biochemical networks³, which exclude real-world metabolic networks. "?

4) How sensitive is the condition to uncertainties in the metabolic network, such as missing or wrong reaction arrows? The number of such possible errors increases with the size of the network, which raises the question regarding the sensitivity of the invariant deficiency, since it is a global condition for the system. Can there be effective decoupling of subnetworks?

Presentation:

5) The manuscript introduces the basic concept of CRNT and states the main result without providing any intuition about the underlying math. The details required for the proof appear only in a rigorous SI. Since CRNT is based on rather elementary properties of linear operators, intuitive argument may be accessible to readers with basic mathematical education.

6) In the format of Nature Communications the authors have enough space for further details. For example, they can specify the stochastic matrix and show its rank for the examples discussed.

7) The title is not so clear: every network has "structure". The authors discuss conditions on the structure of chemical networks which are necessary for concentration robustness.

8) The review by Gunawardena "Chemical reaction network theory for in-silico biologists" (2003) provides much intuition about the underlying math. This work is used in the SI and should be also mentioned in the main text.

-

Reviewer #1 (Remarks to the Author)

Dear Editors,

The manuscript by Eloundou-Mbebi et al follows up on influential recent work by Shinar and Feinberg (among others) to establish structural conditions for absolute concentration robustness (ACR), a condition in which one protein in the network has the same steady state concentration regardless of the total amount of all proteins in the system. Rather than finding sufficient conditions for ACR, as has been the norm in the literature, they set out to find **necessary** structural conditions for this property. They find that in order for a protein to have ACR, its removal cannot change the deficiency of the network. The deficiency is an integer number that is independent of rate constants, i.e. determined exclusively by the network structure.

We would like to thank the reviewer for acknowledging the relevance of the topic, commenting on the timeliness of the work, and for indicating the novelty of the findings. We would like to emphasize that we empirically study the property of ACR in metabolic networks, rather than signaling networks (involving proteins as species); therefore, the species in our networks are metabolites. In our opinion, it is this consideration of metabolic networks together with the applicability of our theoretical results to large-scale real-world metabolic networks that render the work relevant and of interest to the general public.

From a theoretical standpoint this is quite a nice result that complements existing work. As in the work by Shinar and Feinberg, the new condition can be easily determined by just writing the two networks and doing some simple computations in your head. This result could in principle be used to rule out ACR in given proteins assuming knowledge of the relevant network. The authors also carry out some analysis of several metabolic networks, showing that around 15% of the proteins in each system satisfy their necessary condition.

The idea of removing a species from a chemical reaction network is central to this work and it has been addressed in the literature before, see eg the paper by Joshi and Shiu (J Mathematical Chemistry 51(1):153-178, 2013). The authors should include this reference and how their ideas fit into this context.

We thank the reviewer for this observation and suggestion for including the mentioned study, of which we were previously unaware. Indeed, we now see that the idea of removing a species from a chemical reaction network has been tackled in the paper of Joshi and Shiu (J Mathematical Chemistry 51(1):153-178, 2013). In this study, the authors showed that for a given set of rate constants, if a subnetwork obtained upon removal of a species from a given network admits multiple non-degenerated positive steady states, so does the original network. They use this result to identify subnetworks conferring multistationarity to the entire network. In our case, by assuming the existence of a species with ACR, the existence of a positive steady state in the original network implies the existence of a positive steady state in the subnetwork obtained by removal of the ACR species. Therefore, the direction of the argument is opposite (see the proof of well-definedness of the function in Lemma 2.2).

Moreover, our idea was to establish a connection between network invariants, such as structural deficiency, and ACR for a particular species. Therefore, while we use the same network operation as Joshi and Shiu, namely removal of species, our results are of different flavor that allows applications to large-scale networks, typically arising in the study of metabolism. We hope the reviewer will recognize the potential for applications of this result, particularly for planning of targeted experiments which involved costly measuring technologies (as elaborated in the updated version of the introduction and the response to a comment, below).

My main concern with this paper is whether this is the appropriate venue for its publication. This result is not as readily applicable to biologists, since it doesn't actually establish ACR but only narrows possible ACR down to a smaller subset of proteins. Simple titration experiments can also rule out ACR by showing variability in steady state concentrations, even when the network is not known. Also, in my opinion the current quality of the writing and the figures, and the strength of the main result itself, are not up to the standards of Nature Communications.

To address this very important comment, we provide the following arguments: Chemical reaction network theory (CRNT) has provided seminal results which set solid mathematical explanation for relevant biochemical properties and applications (e.g. multistationarity, identifiability). However, many of these results have either been illustrated on small toy networks or are applicable to special classes of networks, which often exclude real-world networks (please, see our response to Reviewer 2, together with a

set of results supporting this claim, now appearing in the updated version of the manuscript). We would like to stress that our results, providing a necessary condition for ACR, offers the possibility to expose this beautiful theory to the more general public, particularly experimental biologists.

As to the applicability of our results for experimental biologists, we offer the following two points:

1. Many results in CRNT preclude the existence of a particular property (for instance, deficiency zero and one theorems are prime examples precluding the possibility to multistationarity). Therefore, having a necessary condition precluding ACR—in a network of any deficiency—is of similar interest and applicability. Moreover, many of the existing results, including those for ACR, provide only a particular sufficient condition for a property (rather than a characterization of the property), which does not render these results any less important.

2. To the best of our knowledge, in molecular biology, detecting metabolites with ACR is not as simple as a titration experiment. One requires mass spectrometric methods with very expensive authentic chemical standards (which often have to be chemically synthesized) to determine the concentration in a given cell material. In addition, the pool of a given metabolite is often divided into several cellular compartments, which imposes additional requirements for measuring subcellular concentration with even more involved techniques (e.g. non-aqueous fractionation). Hence, having carefully planned experiments, to allow resolving concentration on subcellular level, together with availability of expensive measuring standards renders our findings relevant to biologists. Finally, to reveal ACR, the experiment has to be repeated under multiple environments, which tremendously increases the costs of such an experiment with actual findings relevant to biologists. Therefore, having predicted that only a fraction of metabolites should be measured under multiple environments, we are convinced that our findings provide a great improvement in terms of experiment and resource planning (as we are currently doing in our lab).

I did like the theoretical aspects of this work quite a bit, which will be reflected in my comments and suggestions. My opinion is that the authors should submit to a more theoretical venue and work on the quality of the presentation as suggested. In particular, I suspect the main result can be significantly strengthened by eliminating assumptions such as positive deficiency or that deleting the ACR variable preserves the number of reactions. The main text has no structure eg introduction, background, main results, conclusion. It starts defining technical concepts on the second paragraph, which is too early. An introductory section should be expanded, including for instance what the system is robust to (eg with respect to what). A conclusion section should also be included.

We thank again the reviewer for acknowledging the theoretical aspects of the work. The reasons why we are convinced that they are of interest beyond the mathematical community is the applicability of the results to large-scale real-world metabolic networks. Prompted by the comment of Reviewer 2, we also conducted additional set of analyses which pinpoint the relevance of the findings to the biological community.

With respect to strengthening the theoretical findings, we acknowledge the reviewer's statement that the requirement for positive deficiency can be relaxed. As for the requirement for maintaining the number of reactions unchanged, we agree with the reviewer, and note that this was done solely for the purpose of remaining true to the definition of a chemical reaction system provided by Feinberg in his seminal notes. The number of reactions is changed either if a reaction has the same reactant and product (which has no effect on the dynamics, as the reviewer pointed out) or if the removal leads to a duplicate reaction (which is, nevertheless, associated different rate constants and, in this case, affects the dynamics of the system). We consider this update in refining the analysis of the large-scale networks we initially considered in our analysis.

Finally, we would like to thank the reviewer for encouraging us to restructure and expand the manuscript. The present style is due to the direct transfer of the manuscript from *Nature*. In the updated version, we have also addressed the general points of robustness, and have improved the quality of the figures (following the reviewer's specific comments, below).

Specific comments:

* page 2, top: 'which lies'

Corrected.

* page 2, top paragraph: A simple subfigure with a sample reaction and its corresponding stoichiometry matrix would clarify these concepts.

We thank the reviewer for suggesting direct inclusion of a subpanel. In the revised version, we have provided such an alteration to Figure 1.

* page 2, 'The set of nodes which are...': This sentence is confusing - if I didn't already know what a linkage class is, I don't think I would understand from reading this.

Thanks for raising this concern. We modified the indicated sentence accordingly and provided further illustration on Figure 1.

* Main Theorem: Isn't the theorem true also for systems with zero structural deficiency? In that case the conclusion must be true, since the deficiency after eliminating S must also be zero. I didn't see that you use this assumption anywhere in the proof.

We thank the reviewer for raising the concern. Indeed, the theorem remains true for deficiency zero networks. Our previous, inconsistent wording was due to misinterpretation of the result of Feinberg and Shinar, on which we now comment in the updated version of the manuscript.

* Main Theorem: You assume that removal of S does not alter the number of reactions, to prevent reactions of the form $X \rightarrow X$. Such reactions are ruled out in chemical reaction networks, simply for the reason that they would not change the dynamics, eg the reaction vector is $X - X = 0$. I think this assumption is artificially eliminating candidates that could have the ACR property (or that maybe can never have it, you can discuss this too).

We thank the reviewer for this suggestion. As the reviewer alluded, we did include this assumption only so that we do not contradict the definition of a chemical reaction system in the seminal notes of Feinberg. We do see and agree that presence of such reactions do not alter the species formation function. Therefore, we included a remark in the Supplementary information and have updated the statement of the theorem accordingly. We updated the results presented in Figure 2 and the Supplementary Table 2 accordingly.

* page 3, bottom paragraph: what do you mean by 'characterizing the energy status of biological systems'?

By 'characterizing the energy status of biological systems', we mean the compounds which provided the energy driving the biochemical reactions in metabolic networks. These compounds are well-defined, and it is their biochemical role which allowed us to single them out in investigating the possibility for ACR. For simplicity, we removed the mention of deoxy-adenosine phosphates.

* page 4, 'In contrast to archea...': In the first sentence of this paragraph you say that certain compounds in animals satisfy ACR necessary condition. In the next sentence you give as example molecules that do not satisfy ACR. Please clarify.

We thank the reviewer for this important point. Due to updates to the main text, this sentence no longer appears in the updated version of the manuscript.

* page 4, bottom: 'The network upon removal' -> 'The network in panel (a) upon removal'

Corrected as suggested.

Supplementary Material Comments

* Author descriptions: Georg Basler is indentified by number 3) in the main manuscript, here with 2).

Corrected as suggested.

* Def 1.1, 2) "of distinct vectors $y \in \dots$ "

Corrected as suggested.

* Def 1.1 3) "where $(y,y') \in R$ is denoted by $y \rightarrow y'$, and we say y 'reacts to' y' . Also, we assume that... "

Corrected, as suggested.

* Def 1.4, bottom: c_y should be denoted c^y .

We thank the reviewer for this remark. However, we that the notation we used is more convenient to avoid confusion with $c_j^{y_j}$, which in this case, corresponds to the concentration of species j to the power of its stoichiometric coefficient in complex y . We hope the reviewer agrees with our decision about this notation.

* Def 1.5: Are you assuming that the number of reactions is not changed upon removal of S? If so, you should state so explicitly throughout the manuscript.

We thank the reviewer for this remark. Following the arguments we provided above, and the remark appearing after the proof of the theorem in the Supplementary material, we find this change obsolete.

* After (5): A_k is also a linear map, not non-linear as you state

We thank the reviewer for this remark. However, since x_y is defined as the product of the components of x to the power of their multiplicity in y , the function A_k is non-linear in x . We suspect that the reviewer means that A_k is linear in x_y . Therefore, we do not see a problem with the definition, which is also credited to Gunawardena (reference 7 in the Supplementary information, now included in the main text as suggested by Reviewer 2).

* after (6): define w_y

The definition appears in the line following (7).

* Remark 1.14: You can include a proof here after what you wrote above - it is obvious that $\text{Im } A_k \subseteq Q$, $\text{sod}_d = \dim \ker Y | \text{Im}(A) \leq \dim \ker Y | Q = d_s$

We thank the reviewer for this remark. The proof to the Remark 1.14 has been added in the updated version of the Supplementary information.

* page 7, item 2.: We show AN isomorphism between...

Corrected, as suggested.

* page 7, last displayed equation: this relation notation seems awkward - why not just label the species S_1 through S_m , and refer to them as S_i, S_j etc?

We thank the reviewer for this remark. However, we used this notation to emphasize the fact that the set of species is ordered.

* page 7, before prop 2.1: Again, why nonzero deficiency? In the zero deficiency case it would appear that the theorem also holds, because the necessary condition is always satisfied.

Please, see the response on the same issue, above.

* Proposition 2.1: This proposition is referred to as prop 3.1 everywhere below. Same problem with the lemmas etc. You should use floating proposition labels in case you change section numbers

We thank the reviewer for this remark. We opted to keep the floating point for easy reference of the appearance of the results.

* Lemma 2.2: this statement is very unclear - you state that a certain network is 'a reaction system', and then you proceed to define rate constants in a very specific way. The rate constants you choose are your own choice, and they don't prove that the network is well defined. Instead of Lemma 2.2, you should write a definition of the rate constants as you want to do it, and then show that this definition has desired properties.

Corrected, as suggested.

* equation (16): sum over y in C_{-S} ?

This equation refers to the reaction system T , so c_{-S} is not needed here.

* page 9, top: What is the D here, presumably the result of deleting S ?

We thank the reviewer for this remark. As mentioned by the reviewer, D is the result of deleting S . It is defined in the statement of what is now Lemma 2.2 in the updated version of the manuscript.

* eq (19): I don't understand this notation Y_S . You can without loss of generality assume that the ACR species is eg the first one S_1 or the last one S_m , and simplify the notation throughout.

Y^S_{\cdot} is the row corresponding to the species S in the matrix Y .

* Proposition 2.6: "and such that removal of S doesn't change the number of reactions..."

We thank the reviewer for this remark. In light of the other changes made above, this does not longer appears in the updated version of the manuscript.

* Lemma 2.7: why is it that you didn't need to deal with duplicated complexes in the previous analysis?

We did not have to deal with the duplicated complexes since the transformation used before operated on linear combination of complex concentrations. Lemma 2.7 is now Lemma 2.6 in the updated version of the manuscript.

* Lemma 2.7, eq (29): script Q

Corrected, as suggested.

* Lemma 2.7, after (29): "where M is the matrix that removes the rows... "

We thank the reviewer for this remark. Corrected, as suggested.

* Lemma 2.7: It is unclear if Q refers here to the set for the original reaction or for the reaction after removing S

See Proposition 1.13. for further specification.

* Theorem 2.8: you never seem to have used above that deficiency >0 for the original system. You did assume that removal of S did not alter reaction, but you didn't write this explicitly in the propositions.

Addressed as the other deficiency zero points.

Reviewer #2 (Remarks to the Author):

1) How strong is the condition in real networks? In other words, can the authors estimate, e.g. by some random sampling:

We would like to thank the reviewer for this very interesting question and suggestion. We provide detailed responses below.

a. What is the fraction of species that obey ACR but their omission changes the number of reactions, and are therefore outside the scope of the condition?

Answering this question requires that either (i) we know the species that are ACR or (ii) we have sufficient conditions allowing to identify species with ACR. In both situations, sampling of the flux space (and determination of respective concentration with a given specific set of rate constants) cannot be used to determine whether a species is ACR. The reason for this is the fact that ACR property holds (i) for any environments, hence, any initial conditions, and (ii) for any choice of rate constants resulting in a positive steady state. Nevertheless, such sampling procedure can be used to rule out that a particular species has ACR. This is precisely where one would benefit from our theoretical findings: We establish necessary conditions that, once violated, rule out the possibility for ACR. The strength of the result is that we do not have to rely on any tedious (and perhaps biased) sampling strategy.

In addition, following a suggestion from Reviewer 1, we updated the theorem so that our condition of equality of structural deficiencies with and without a particular species holds irrespective of the change in the number of reactions.

b. What is the fraction of species that obey the condition of invariance of deficiency and number of reactions, but do not exhibit ACR?

As mentioned above, sampling would not provide the means for addressing this question, as it would hold only for the tested set of rate constants and initial conditions (which is no guarantee for ACR). Therefore, no strong statement could be made as an answer to the interesting point of the reviewer.

2) How general is the present necessary condition with respect to the previous sufficient condition of Shinar and Feinberg:

a. How many of the ACR cases observed by the authors would be missed by the previous condition?

Our theorem provides a necessary condition that is applicable to networks of arbitrary deficiency. The sufficient condition of Shinar and Feinberg is only applicable to networks of deficiency one. It is for this type of networks, under additional conditions, that they can identify ACR species. Any of these species do not violate our condition. Therefore, for deficiency one network, the species that satisfy our condition may include additional ACR species which cannot be identified by the result of Shinar and Feinberg. In addition, for the networks we analyzed, any of the species which satisfy our necessary condition for ACR will not be identified by the approach of Shinar and Feinberg. The reason for this strong result is that all of the networks we analyzed are of deficiencies greater than one (please, see the updated Supplementary Table 2). In this sense, our results can be used for guiding carefully planned metabolomics experiments targeted to establishing concentration robustness for specific metabolites.

b. How many of the ACR cases that obey the Shinar-Feinberg condition violate the present condition?

No such species violates our condition. We have a necessary condition which is satisfied by any ACR species.

3) "... some of these properties cannot be efficiently computed for large-scale biochemical networks^{12,13}, usually employed in studies of metabolism¹⁵⁻¹⁷, as they depend on the rate constants. "

a. the authors should elaborate on what exactly are the properties that cannot be efficiently computed and why.

b. A basic concept of CRNT is that counting dimensions and calculating topological indices such as the deficiency, often provides a lot of information regarding fixed points etc. For example, the Shinar-Feinberg is purely structural. Which CRNT criteria "depend on the rate constants"?

In the following, we provide the answers to items a and b, above. While it is true that calculation of structural deficiency is computationally feasible, allowing for application of the sufficient condition of Shinar and Feinberg to networks of deficiency one, it is the general procedure of Karp et al. (please, see the updated references) that becomes infeasible for networks of larger size (please, see statement on pp. 7 in the reference). The reason lies in the symbolic computation related to establishing complex invariants of type 1. For this reason, Karp et al. devised a heuristic applicable to special type of complex invariants, called "type 1". Since these details are too technical for the general audience, we opted to include them in a much simplified form which will not compromise the understandability by the general audience. We thank the reviewer for raising this point, and we rephrased this part of the manuscript to avoid potential confusion.

As an example of a CRNT criterion which depends on the rate constants, we would like to mention the concept of dynamic deficiency (used in a part of our proof, please, see Supplementary Information). Our first result in fact relies on establishing an isomorphism which implies invariance of dynamic deficiency upon removal of a species with ACR (please, see Proposition 2.1).

c. Can the authors substantiate the claim that the other criteria "are too restrictive for application with real-world networks, as they hold for a very special class of biochemical networks³, which exclude real-world metabolic networks. "?

We thank the reviewer for this comment. We provided additional results demonstrating the deficiency of the networks we investigated in our study. Please, refer to the updated version of Supplementary Table 2 for demonstration that all of the studied networks are of deficiency greater than one. Therefore, the results of Shinar and Feinberg are, indeed, not applicable here, as claimed in the first version of the manuscript.

4) How sensitive is the condition to uncertainties in the metabolic network, such as missing or wrong reaction arrows? The number of such possible errors increases with the size of the network, which raises the question regarding the sensitivity of the invariant deficiency, since it is a global condition for the system. Can there be effective decoupling of subnetworks?

This comment is particularly insightful and addressing it greatly expanded the scope of our study. To examine the sensitivity of our findings to this bias in the network, we considered removing 1, 2, 5, 7, and 10% randomly selected reactions from each of the analyzed network. We quantified the effect of the reaction removal we focus on those metabolites that satisfy the necessary condition in the original network, but happen to violate this condition upon the network perturbation. We refer to such metabolites as switching metabolites. We then determine the switching ratio, defined as the proportion of switching metabolites from those which satisfy the necessary condition in the original network. The switching ratio for every network and perturbation level was determined over at least 40 samples for each considered percentage of removed reactions.

Our results indicate that the switching ratio depends on a small set of reactions (Figure 3 and Supplementary Figure S7). This is supported by our findings that already the removal of 1% of reactions, on average, leads to as large value for the switching ratio as the removal of 10% of the reactions. In addition, the observation that the switching ratio is not larger than 50%, with exception to *Chlamydomonas reinhardtii*, indicates that a certain set of metabolites satisfies the necessary condition even upon all considered levels of perturbations. This may point out that these metabolites may essentially participate in pathways which may be effectively decoupled in the considered networks.

Another source of uncertainty of large-scale metabolic networks is represented by the directionality of the included reactions. While some reactions are known to operate effectively as irreversible, others may change the operating direction preferentially according to cellular conditions (Noor et al., *Bioinformatics* 28, 2037-2044, 2012). Nevertheless, changing the directionality of a reaction does not affect the number of complexes and the number of linkage classes. Moreover, the rank of the stoichiometric matrix is not affected by change of directionality. These observations together with the definition of structural deficiency indicate that change of reaction directionality does not affect the structural deficiency. Therefore, our results are not affected by possible uncertainty in reaction directionality.

We added two paragraphs to provide an explanation of these points in the updated version of the manuscript.

Presentation:

5) The manuscript introduces the basic concept of CRNT and states the main result without providing any intuition about the underlying math. The details required for the proof appear only in a rigorous SI. Since CRNT is based on rather elementary properties of linear operators, intuitive argument may be accessible to readers with basic mathematical education.

We have added an additional paragraph to the main text explaining in an intuitive way the essence of the approach taken. We hope that the reviewer recognizes the improvements of the presentation in the updated version of the manuscript.

6) In the format of Nature Communications the authors have enough space for further details. For example, they can specify the stochastic matrix and show its rank for the examples discussed.

Reviewer 1 had a similar suggestion. We have included an additional panel for illustrating the key concepts. We thank the reviewer for this request.

7) The title is not so clear: every network has "structure". The authors discuss conditions on the structure of chemical networks which are necessary for concentration robustness.

We thank the reviewer for this remark. We opted to change the title to "A network property necessary for concentration robustness".

8) The review by Gunawardena "Chemical reaction network theory for in-silico biologists" (2003) provides much intuition about the underlying math. This work is used in the SI and should be also mentioned in the main text. -

The reference has been included in the main text of the updated version of the manuscript.

Reviewer #1 (Remarks to the Author)

Dear Editor,

The authors have adequately answered the majority of my concerns, for instance by numerous changes to the text, a new introduction, and pointing out the difficulty of carrying out titration experiments in the context of metabolic networks. The manuscript has become much stronger as a result, and I thank the authors for their work. However there are still a few issues that need to be addressed before it can be accepted for publication in my opinion.

The authors included several paragraphs of introduction before starting with the technical definitions, which improves the strength and readability of this manuscript. The paper is still not organized with labels such as 'intro', 'conclusions' etc, but it uses labels that are perhaps appropriate for a paper of this short length. The readability was also improved by using examples and figures in their definitions.

I agree with Reviewer 2 that it would be very helpful to know what fraction of species obey the condition but don't exhibit ACR. The authors point out that determining ACR technically would involve testing for infinitely many initial conditions and parameter values. But this can be reasonably be done by using a finite number of initial conditions and parameters. If a variable appears to be ACR using such a test, it can be declared to be 'likely ACR'. Moreover, a number of variables could be found this way that obey the property but are not ACR. For that matter, it would be helpful to show in the supplement a single, simple example of a system that satisfies the condition but is not ACR.

A numerical analysis such as described above could also address the issue of when absolute robustness is 'vacuously true' in the sense that there are no mass conservation laws whatsoever, or more precisely, there is a unique steady state in the system regardless of initial condition. Presumably a fraction of the systems found satisfy ACR for this reason, which is not particularly interesting.

I think your position is still unclear dealing with the case where there are reactions such as $y \rightarrow y + S$. The removal of S leads to $y \rightarrow y$ which doesn't satisfy the traditional definitions. If you want to include such systems in your analysis, you need to change Definition 1.1 #3 and make sure that you can still do the analysis under the new definition (and be careful when you make references to the literature). If you don't want to include such systems, then you still need to rule them out in the assumptions of all relevant results.

Regarding the notation c_y in def 1.4: OK to do this but then you need to be consistent - for example in equation (14) you write c^y and after the equation define c^y instead of c_y . Then you use the latter notation many times in the subsequent text.

Reviewer #2 (Remarks to the Author)

The authors made a serious effort to improve the original manuscript. In the revised version, they removed certain limitations from their main theorem, which allows them to better compare its utility with respect to the more specific, but stronger, result by Shinar and Feinberg. They addressed one concern regarding the sensitivity of the result to uncertainties and missing information about the structure of metabolic networks. At the same time, they explain quite frankly why other concerns cannot be addressed by the present approach. Overall, I feel that the theoretical result, which by itself is rather elementary, is of interest to the computational biology community, mainly for its practical use in actual metabolic networks. I therefore support the consideration of the manuscript for publication, after the authors make additional effort to clarify the presentation.

Reviewer #1 (Remarks to the Author):

Dear Editor,

The authors have adequately answered the majority of my concerns, for instance by numerous changes to the text, a new introduction, and pointing out the difficulty of carrying out titration experiments in the context of metabolic networks. The manuscript has become much stronger as a result, and I thank the authors for their work. However there are still a few issues that need to be addressed before it can be accepted for publication in my opinion.

The authors included several paragraphs of introduction before starting with the technical definitions, which improves the strength and readability of this manuscript. The paper is still not organized with labels such as 'intro', 'conclusions' etc, but it uses labels that are perhaps appropriate for a paper of this short length. The readability was also improved by using examples and figures in their definitions.

We thank the reviewer for acknowledging the substantial changes to the manuscript thanks to the earlier insightful comments. While preparing the manuscript, we followed the guidelines for an Article format of *Nature communications* (from the webpage http://www.nature.com/ncomms/authors/content_types.html “The main text of an Article should begin with an introduction (without heading) of referenced text that expands on the background of the work (some overlap with the abstract is acceptable), followed by sections headed Results, Discussion (if appropriate) and Methods (if appropriate). The Results and Methods sections may be divided by topical subheadings; the Discussion should be succinct and may not contain subheadings.”). In the updated version, we further streamlined the text and addressed the remaining points.

I agree with Reviewer 2 that it would be very helpful to know what fraction of species obey the condition but don't exhibit ACR. The authors point out that determining ACR technically would

involve testing for infinitely many initial conditions and parameter values. But this can be reasonably be done by using a finite number of initial conditions and parameters. If a variable appears to be ACR using such a test, it can be declared to be 'likely ACR'. Moreover, a number of variables could be found this way that obey the property but are not ACR. For that matter, it would be helpful to show in the supplement a single, simple example of a system that satisfies the condition but is not ACR.

A numerical analysis such as described above could also address the issue of when absolute robustness is 'vacuously true' in the sense that there are no mass conservation laws whatsoever, or more precisely, there is a unique steady state in the system regardless of initial condition. Presumably a fraction of the systems found satisfy ACR for this reason, which is not particularly interesting.

We thank the reviewer for raising this point:

First, we followed the suggestion and expanded the section of the example in Supplementary Discussion (Section 3 of the Supplementary information). For this example we have an analytical solution for the steady-state concentrations (please, refer to Eq. (41) therein). Therefore, we state that "However, there exist some species, such as: A, D, E, and F, whose removal does not alter the deficiency of the original network but are not ACR (see analytical solution below). For such species, our theorem remains silent." In addition, we updated the main text to point to this example, indicating the components of the system that satisfies the condition but is not ACR.

Second, we used a medium-scale kinetic model of *E. coli* consisting of 830 components (metabolites and enzymes) and 1330 reactions. The kinetics of reaction rates follows mass action, and we ensured the used positive rate constants yield a positive steady state from the initial condition provided in the original publication (Khodayari *et al.* (2014) *Metabolic engineering* 25). Next, we used 150 different initial conditions also leading to positive steady states. We then checked for which components the steady-state concentration was invariant with respect to the changes in the inspected initial conditions. In addition, for this network we also determined the components which satisfy our condition (of leaving the deficiency unchanged upon the component removal). Altogether, we identified 2 components which are 'likely' ACR as they satisfy our network property necessary for ACR and are of invariant concentration with respect to the inspected 150 initial conditions (see Supplementary Table 3). We added a paragraph on these findings in the main text and expanded the methods and the Supplementary information accordingly.

We agree with the reviewer that an example in which there is a single positive steady state regardless of the initial condition is not interesting. The network in the example of Section

3 of the Supplementary information is mass conservative, in the sense that there exists a positive vector $w(1, 2, 2, 4, 3, 4)$, for which $w(y'-y) = 0$ for all reactions $y \rightarrow y'$. Moreover, in using the real-world networks, we ensured that they are of high quality, i.e. they preserve element and charge balances; hence, they conserve mass and can be used to study cases that do not vacuously hold.

I think your position is still unclear dealing with the case where there are reactions such as $y \rightarrow y + S$. The removal of S leads to $y \rightarrow y$ which doesn't satisfy the traditional definitions. If you want to include such systems in your analysis, you need to change Definition 1.1 #3 and make sure that you can still do the analysis under the new definition (and be careful when you make references to the literature). If you don't want to include such systems, then you still need to rule them out in the assumptions of all relevant results.

According to Feinberg, whenever $y \rightarrow y$ is in the network, it should be removed (in order to work with a well-defined chemical reaction network), as it does not affect the dynamic of the system. Therefore, we added remark (1.2) to emphasize this point and we rephrased remark (1.7) in the Supplementary information, accordingly.

Regarding the notation c_y in def 1.4: OK to do this but then you need to be consistent - for example in equation (14) you write c^y and after the equation define c^y instead of c_y . Then you use the latter notation many times in the subsequent text.

We agree with this suggestion, and we made the change in notation throughout the Supplementary information.

Reviewer #2 (Remarks to the Author):

The authors made a serious effort to improve the original manuscript. In the revised version, they removed certain limitations from their main theorem, which allows them to better compare its utility with respect to the more specific, but stronger, result by Shinar and Feinberg. They addressed one concern regarding the sensitivity of the result to uncertainties and missing information about the structure of metabolic networks. At the same time, they explain quite frankly why other concerns cannot be addressed by the present approach. Overall, I feel that the theoretical result, which by itself is rather elementary, is of interest to the computational biology community, mainly for its practical use in actual metabolic networks. I therefore support the consideration of the manuscript for publication, after the authors make additional effort to clarify the presentation.

We thank the reviewer for acknowledging the substantial changes to the manuscript thanks to the earlier insightful comments. In the updated version of the manuscript, we

attempted to highlight some additional examples from the Supplementary information, provided another simulation study to check for likely ACR components in a medium-size kinetic model of *E. coli* (Khodayari *et al.* (2014) *Metabolic engineering* 25), and streamlined the text to further clarify and improve presentation.

By conducting this additional analysis, we identified 2 components which are ‘likely’ ACR as they satisfy our network property necessary for ACR and are of invariant concentration with respect to the inspected 150 initial conditions (see Supplementary Table 3). We added a paragraph on these findings in the main text and expanded the methods and the Supplementary information accordingly.